# The Citrus Mutant *Jedae-unshiu* Induced by Gamma Irradiation Exhibits a Unique Fruit Shape and Increased Flavonoid Content

**DOI:** 10.3390/plants11101337

**Published:** 2022-05-18

**Authors:** Chang-Ho Eun, In-Jung Kim

**Affiliations:** 1Subtropical Horticulture Research Institute, Jeju National University, Jeju-si 63243, Korea; mong6908@jejunu.ac.kr; 2Faculty of Biotechnology, College of Applied Life Sciences, Research Institute for Subtropical Agriculture and Biotechnology, SARI, Jeju National University, Jeju-si 63243, Korea

**Keywords:** *Citrus unshiu*, irradiation, *Jedae-unshiu*, flavonoid, epidermal cells

## Abstract

Satsuma mandarin (*Citrus unshiu* Marc. cv. Miyagawa-wase) is the most widely cultivated citrus variety in Korea. Although most varieties are imported from Japan, efforts have focused on developing new domestic varieties. We produced mutants by irradiating *C. unshiu* Marc. cv. Miyagawa-wase scions with gamma rays and grafting them onto *C. unshiu* Marc. cv. Miyagawa-wase branches. We compared the characteristics of these mutants with Miyagawa-wase fruit as a control. A mutant line named *Jedae-unshiu* with a unique fruit shape was selected for investigation in detail. The phenotype of *Jedae-unshiu* fruit demonstrated vertical troughs on the flavedo, smooth albedo without rough protruding fibers, and good adhesion between peel and flesh. In addition, *Jedae-unshiu* had thicker peels and higher fruit hardness than the control. Higher levels of hesperetin and narirutin, representative flavonoids, accumulated in the peel and flesh of *Jedae-unshiu* than those of the control. Cellular-level microscopic observations of the mature fruit peels demonstrated epidermal cell disruption in the control but not in *Jedae-unshiu*. Our results suggest that *Jedae-unshiu* has high possibility for development as a good storage variety containing large amounts of flavonoids, in addition to potential for ornamental value due to the unique shape of the fruit.

## 1. Introduction

Citrus is a representative crop that ranks first or second among fruit crops in terms of production value in Korea, and Jeju Island in Korea is the main production area. However, cultivation is gradually increasing in southern regions, such as Jeollanam-do, due to recent climate warming. The goal of citrus breeding in western countries, including the United States, is to develop mandarin varieties that have no seeds and are easy to peel, such as *Citrus unshiu*. Early-maturing satsuma mandarin (*C. unshiu* Marc. cv. Miyagawa-wase) is the most widely cultivated variety of citrus on Jeju Island. Miyagawa wase originated in Fukuoka Prefecture of Japan as a natural mutant of ordinary *C. unshiu*. This strain was later introduced into Korea by a French priest in 1911 and cultivated as a representative variety. *C. unshiu* Marc. var. Okitsu, the second-most cultivated citrus variety on Jeju Island, is derived from a branch mutation of *C. unshiu* Marc. cv. Miyagawa-wase. The cultivation area of *C. unshiu* Marc. cv. Miyagawa-wase accounts for more than 80% of the total citrus cultivation area on Jeju Island. Most produced citrus fruits are consumed as raw food, and non-commercial-sized fruits are processed into products, such as juice. It is necessary to develop a variety that can not only be consumed raw but also cultivated for ornamental purposes and processed with enhanced functionality [1,2,3,4,5]. The improvement of citrus varieties through traditional crosses is difficult due to inherent biological factors, such as apomixis, self- and cross-incompatibility, long seedling periods, and high heterozygosity [6,7,8]. Several single-embryonic varieties of the mandarin lineage exist, which facilitate traditional breeding, although mandarin breeding by artificial mutagenesis is also performed. Seeds are not typically formed by *C. unshiu* Marc. cv. Miyagawa-wase, which was used in this study, but seeds can be obtained through artificial crossbreeding with the seed-forming maternal plant. However, because the seeds are polyembryonic and little pollen is formed, the development of new varieties through crossbreeding is difficult. Gamma irradiation, a mutagenesis method used to develop new varieties, is an important component of citrus breeding programs and is mainly used to improve varieties of grapefruit, pomelo, lemon, and mandarins [9,10,11,12]. In this study, *C. unshiu* Marc. cv. Miyagawa-wase scions were irradiated with gamma rays and then grafted onto the branches of *C. unshiu* Marc. cv. Miyagawa-wase trees. Mutant lines indicating differences in specific shape, time to maturation, sugar content, and skin color were selected by comparing the fruit produced by grafted branches with control fruit. Among them, a mutant called *Jedae-unshiu* presented a unique fruit shape. The external characteristics, sugar, acidity, skin color, flavonoid content, and peel structure at the cellular level were investigated in this mutant fruit.

## 2. Results and Discussion

### 2.1. Selection of Mutant Lines Produced by Gamma Irradiation

We previously performed a gamma ray irradiation (^60^Co, 100Gy with 60% survival rate of budsticks) onto *C. unshiu* Marc. cv. Miyagawa-wase scions and then grafted onto the branches of *C. unshiu* Marc. cv. Miyagawa-wase trees to select mutant lines. Several mutant lines, such as a unique fruit shape, high sugar content, peel color, or late maturation, etc. were selected that year. The next year, the scions of mutant lines selected were used for veneer-grafting onto rootstocks (*C. unshiu* Marc. cv. Miyagawa-wase) and the lines were maintained (Figure 1) [13,14]. Of several mutant lines, *Jedae-unshiu*, demonstrating a unique fruit shape, was selected for further analysis and investigated for whether its traits were maintained over 3 years, from 2018 to 2020.

### 2.2. Comparison of External Morphological Traits between Jedae-unshiu and Control Fruits

The external morphological differences between control and *Jedae-unshiu* fruits induced by irradiation were investigated. From immature fruit, the outer colored peel layer (flavedo) of *Jedae-unshiu* fruit demonstrated vertical troughs that were absent in control fruit (Figure 2A,B), and the white inner peel layer (albedo) was smooth, lacking the rough protruding fibers observed in the control (Figure 2C). In addition, *Jedae-unshiu* fruits had good adhesion between the peel and flesh and did not differ from the control fruit in terms of peeling. In a tomato mutant induced by gamma irradiation, the fruit was abnormally developed and peanut-shaped, unlike the round fruit of the control [15]. Ryu et al. (2020) reported that strawberry plants irradiated with gamma rays produced fruits that were smaller or lacked pigment [16].

In a comparative analysis between *Jedae-unshiu* and control fruits, no significant differences in horizontal length, vertical length, or fruit weights were observed in the 3 years from 2018 to 2020. However, over these 3 years, the average peel thickness was 2.30 ± 0.30 mm for the control, compared with 3.05 ± 0.65 mm for *Jedae-unshiu*, indicating that *Jedae-unshiu* had thicker peels than the control. *Jedae-unshiu* also had a higher fruit hardness than the control (1016.50 ± 111.48 g versus 710.01 ± 112.68 g; Table 1). We confirmed that the peel thickness and fruit hardness traits of *Jedae-unshiu* were maintained for 3 years. No significant differences were found between *Jedae-unshiu* and the control in fruit sugar contents, acidity, or color (Table 2).

### 2.3. Comparison of Flavonoid Compounds in Jedae-unshiu and Control Fruits

Flavonoids were extracted from the peel and pulp of mature *Jedae-unshiu* and control fruits, and flavonoid component analysis was performed by HPLC. Lutin, naringin, neohesperidin, hesperidin, narirutin, naringenin, and hesperetin were used as standard materials. As indicated in Table 3, the highest flavonoid components of mature *Jedae-unshiu* and control fruits were hesperetin and narirutin. Nogata et al. (2006) also reported that hesperidin is the most abundant flavonoid in the peels of satsuma mandarin (*C. unshiu*) [17]. According to the quantitative analysis, the hesperetin and narirutin contents in the peel of *Jedae-unshiu* were 546.6 ± 3.38 mg and 78.5 ± 0.61 mg per 100 g of dry weight, respectively, compared with 463.1 ± 1.11 mg and 60.8 ± 1.51 mg per 100 g of dry weight, respectively, in the peel of the control. In the flesh, the hesperetin and narirutin contents in *Jedae-unshiu* were 276.3 ± 2.07 mg and 132.2 ± 1.53 mg per 100 g of dry weight, respectively, compared with 178.8 ± 0.43 mg and 92.4 ± 0.59 mg per 100 g of dry weight, respectively, in the control. The fruit of *Jedae-unshiu* had a higher flavonoid content than that of the control (Table 3). Oufedjikh et al. (1998) investigated changes in flavonoid contents during storage at 3 °C after irradiating *Moroccan clementine* fruit with gamma rays and found significantly higher hesperidin, nobiletin, and heptamethoxyflavone contents in irradiated fruit than in the control fruit after 14 days of storage [18]. Moghaddam et al. (2011) reported an increase in the total flavonoid contents of *Centella asiatica* irradiated with gamma rays compared with control fruit [19]. In paprika (*Capsicum annuum* L.), the abundance of phytochemicals other than flavonoids, including capsaicinoids, increased following irradiation with gamma rays [20].

### 2.4. Cell-Level Microscopic Observations of Peels from Jedae-unshiu and Control Fruits

The peel structures of immature (harvested in June) and mature (harvested in December) *Jedae-unshiu* and control fruits were compared at the cellular level by microscopic examination (Figure 3). No significant differences in morphological characteristics were observed at the cellular level between the peels of immature *Jedae-unshiu* and control fruits. However, in the peels from mature fruits, epidermal cell disruption was observed in control but not in *Jedae-unshiu* fruits, suggesting a delay in epidermal cell breakdown in mature *Jedae-unshiu* compared with control fruit. In addition, in the peel of mature *Jedae-unshiu*, the distribution of vascular bundles favored the longitudinally protruding portion of the peel over the flat portion, whereas the flat portion of the peel had a more developed intercellular space compared with the protruding portion (data not displayed).

In summary, a new satsuma mandarin mutant (*Jedae-unshiu*) was developed by exposing *C. unshiu* Marc. cv. Miyagawa-wase to gamma irradiation. The phenotype of *Jedae-unshiu* fruit included vertical troughs on the flavedo, a smooth albedo without rough protruding fibers, and good adhesion between the peel and the flesh. *Jedae-unshiu* had a thicker peel and higher hardness compared with the control. The contents of hesperetin and narirutin, which are representative flavonoids found in satsuma mandarin, were higher in *Jedae-unshiu* fruit than in control fruit. The breakdown of the epidermal cells in the *Jedae-unshiu* peel was delayed compared with the breakdown in the control peel. Taken together, these findings suggest that *Jedae-unshiu* has high potential for development as a storable variety containing high levels of flavonoids and great value for ornamental use due to the unique shape of the fruit.

## 3. Materials and Methods

### 3.1. Chemicals

Methanol, ethanol, acetonitrile, acetic acid, formalin, and tert-butyl alcohol were purchased from Fisher Scientific (Seoul, Korea). Rutin, naringin, neohesperidin, hesperidin, narirutin, naringenin, hesperetin, crystal violet, xylene, safranin, and fast green were purchased from Sigma-Aldrich (St. Louis, MI, USA).

### 3.2. Plant Materials

Young and mature fruits of *C. unshiu* Marc. cv. Miyagawa-wase, as a control, and its mutant line *Jedae-unshiu* were used in this study.

### 3.3. Analysis of Fruit Traits

Fruit weight was measured using an electronic indicator scale (CAS Co., Ltd. YangJu, Korea). The vertical diameter, transverse diameter, and peel thickness of the fruit were measured using a digital caliper (MITUTOYO Corporation, Kawasaki, Japan). Fruit hardness was measured using a fruit hardness meter (LUTRON FR-5105, Antala Staška, Czech Republic). Changes in the shape of the fruit were observed with the naked eye, and the section of the fruit peel was observed through a microscope (NIKON Corporation, Tokyo, Japan). Sugar content and acidity were measured in 4–5 mL of fruit juice according to the instruction manual for the NH-2000 (HORIBA, Kyoto, Japan). Changes in the fruit peel color were measured using a chromometer (CR-400, MINOLTA, Tokyo, Japan). *Jedae-unshiu* and control were investigated by harvesting 10 or more fruits per tree from 5 or more trees each year.

### 3.4. Detection of Flavonoid Components by High-Performance Liquid Chromatography (HPLC) Analysis

Flavonoids were extracted from freeze-dried fruit according to the extraction method described by Senevirathne et al. [21]. Each freeze-dried sample (peel and pulp) was ground with a grinder and then sifted through a 0.5-mm mesh. A volume of 250 mL of 100% methanol was added to 2.5 g of the powdered sample, and flavonoids were extracted in a shaking incubator at 25 °C for 24 h. After incubation, the supernatant containing flavonoids was filtered through a syringe with a 0.2-µm filter prior to injection into an HPLC system. Rutin, naringin, neohesperidin, hesperidin, narirutin, naringenin, and hesperetin were used as a standard (Sigma, St. Louis, MI, USA).

Flavonoid separation was performed on the Waters (Milford, MA, USA) HPLC system using a Shim-pack VP-ODS(C18) column (4.6 mm × 150 mm inner diameter, 5-µm particle size, SHIMAZU, Kyoto, Japan) in a mobile phase. The chromatographic separation was performed with solvents A (water [99.5%]:acetic acid [0.05%]) and B (acetonitrile [99.5%]:acetic acid [0.05%]) at a flow rate of 1 mL/min. The gradient profile was programmed at 30% B (0–28 min), 30–80% B (28–33 min), 80% B (33–38 min), 80–30% B (38–43 min), and 30–1% B (43–48 min). The column was equilibrated for 10 min with 30% B before the next injection. The sample injection volume was 10 µL, and flavonoids were detected at 280 nm. The results are expressed as the mean ± standard deviation of three replicates. All statistical analyses were carried out using IBM SPSS software (SPSS for Windows, version 20, SPSS Inc., Armonk, NY, USA). Significant differences among the samples were calculated using an analysis of variance followed by Duncan’s multiple range test at the 5% level (*p* < 0.05).

### 3.5. Microscopic Observation of Citrus Peels

The peels of young and mature *Jedae-unshiu* and control fruits that were harvested at the same time were microscopically observed. A 5 mm × 5 mm piece of the outer skin was fixed in formalin–acetic acid (FAA) solution (90 mL of 50% ethanol, 5 mL of formalin, and 5 mL of acetic acid) for 24 h. The fixed tissue was dehydrated and made transparent using the following procedure: 50% ethanol for 5 h; solution I (40 mL of 95% ethanol, 10 mL of tert-butyl alcohol, and 50 mL of distilled water [DW]) for 5 h; solution II (50 mL of 95% ethanol, 20 mL of tert-butyl alcohol, and 30 mL of DW) for 37 h; solution III (50 mL of 95% ethanol, 35 mL of tert-butyl alcohol, and 15 mL of DW) for 5 h; solution IV (45 mL of 55% ethanol, 10 mL of tert-butyl alcohol, and 45 mL of 1% crystal violet) for 18 h; solution V (25 mL of 100% ethanol, 75 mL of tert-butyl alcohol, and 30 mL of DW) for 5 h; and solution VI (100 mL of tert-butyl alcohol) for 5 h. The dehydrated tissues were immediately placed into a carton and covered with paraffin, and the trimmed wax block was fixed on a microtome. The slice thickness was adjusted to 10 µm. The sections were stained as follows: xylene substitute for 60 min and 10 min; a mixture of 100% ethanol and xylene (1:1) for 10 min; 100%, 95%, 85%, and 75% ethanol for 10 min each; 1% safranin for 2 h; 75% ethanol for 5 min; 0.5% fast green for 30 s; 95% and 100% ethanol for 5 min each; a mixture of 100% ethanol and xylene (1:1) for 5 min; and xylene substitute for 10 min twice. The slices were examined and photographed using an optical microscope (Nikon microphoto type 114).

## Figures and Tables

**Figure 1 plants-11-01337-f001:**
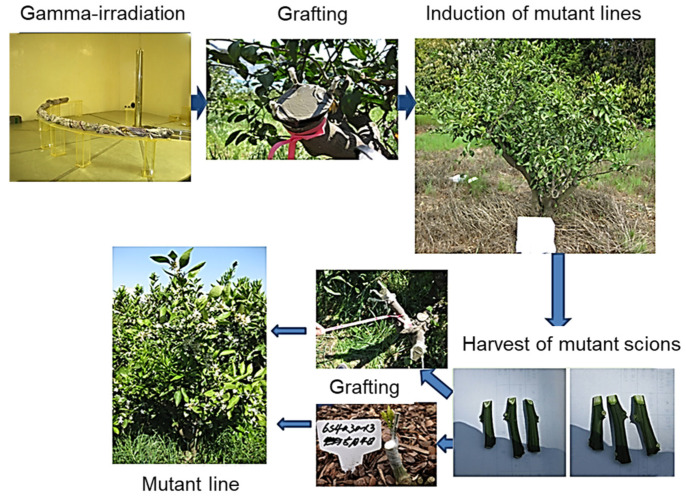
Mutagenesis by gamma irradiation of *C. unshiu* scions and the selection of mutant individuals by grafting.

**Figure 2 plants-11-01337-f002:**
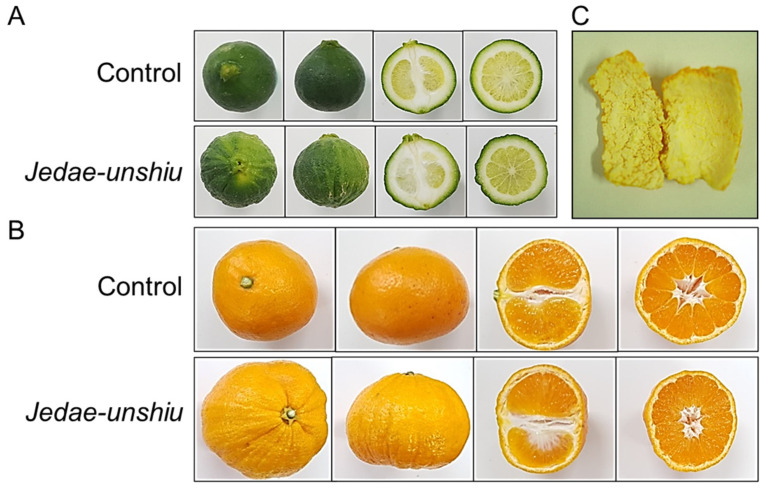
Morphological comparison between *Jedae-unshiu* and control fruits. (**A**) Immature fruits, (**B**) Mature fruits, (**C**) Albedo of control (left) and *Jedae-unshiu* (right).

**Figure 3 plants-11-01337-f003:**
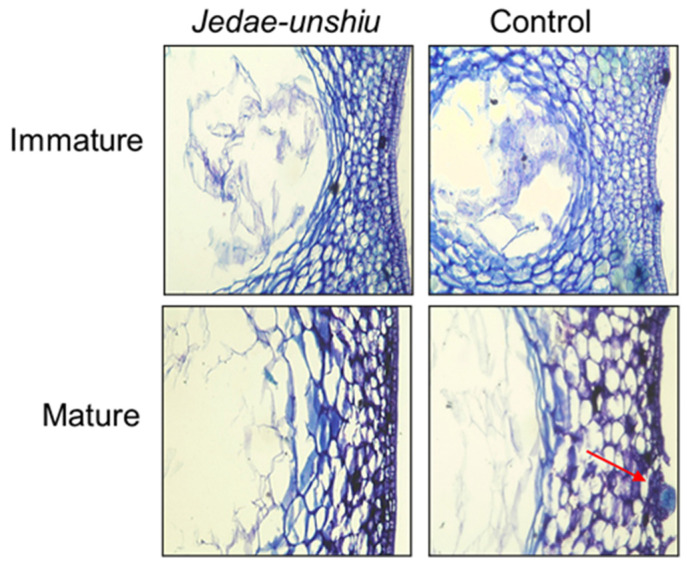
Microscopic observation of immature and mature peels from *Jedae-unshiu* and control fruits. The red arrow indicates the breakdown of epidermal cells.

**Table 1 plants-11-01337-t001:** Comparative analysis of control and *Jedae-unshiu* fruits from 2018 to 2020.

	Year	Horizontal (mm)	Vertical (mm)	Weight (g)	Peel Thickness (mm)	Hardness (G)
Control	2018	52.13 ± 2.36	70.18 ± 2.26	129.36 ± 11.16	2.18 ± 0.37	679.25 ± 130.02
	2019	51.75 ± 6.72	61.03 ± 7.94	97.46 ± 25.92	2.08 ± 0.31	615.90 ± 347.39
	2020	52.65 ± 1.66	61.48 ± 7.80	104.40 ± 7.99	2.65 ± 0.23	834.87 ± 62.47
Average		51.73 ± 0.96	63.58 ± 4.40	105.69 ± 16.63	2.30 ± 0.30	710.01 ± 112.68
*Jedae-unshiu*	2018	61.32 ± 11.18	68.80 ± 9.13	126.55 ± 28.56	3.45 ± 0.58	924.89 ± 379.10
	2019	53.62 ± 5.94	64.75 ± 5.44	109.60 ± 19.72	2.15 ± 0.43	984.00 ± 308.50
	2020	48.46 ± 2.79	61.63 ± 3.72	90.22 ± 11.37	3.01 ± 0.85	1140.62 ± 117.11
Average		53.06 ± 5.99	64.71 ± 3.02	105.60 ± 16.15	3.05 ± 0.65 *	1016.50 ± 111.48 *

* indicates a value with a significant difference compared to the control (*p* < 0.05).

**Table 2 plants-11-01337-t002:** Comparative analysis of sugar, acidity, and hunter color value between control and *Jedae-unshiu* fruits.

	Year	Sugar (Brix)	Acidity (wt%)	Hunter Color Value		
				L	a	b
Control	2018	9.29 ± 0.43	0.65 ± 0.08	61.60 ± 1.65	25.92 ± 1.42	37.16 ± 1.12
	2019	7.35 ± 0.56	0.84 ± 0.09	59.23 ± 3.26	18.54 ± 2.82	35.56 ± 1.95
	2020	7.54 ± 0.79	0.85 ± 0.07	62.38 ± 1.92	24.79 ± 1.17	38.05 ± 0.58
Average		8.28 ± 0.98	0.76 ± 0.10	62.445 ± 3.06	23.65 ± 3.44	44.30 ± 14.80
*Jedae-unshiu*	2018	7.26 ± 0.44	0.68 ± 0.08	59.76 ± 3.46	22.84 ± 2.68	36.27 ± 2.16
	2019	7.76 ± 0.61	0.78 ± 0.06	59.67 ± 3.64	18.51 ± 3.26	36.03 ± 2.68
	2020	7.28 ± 0.68	0.85 ± 0.10	64.45 ± 2.98	22.19 ± 2.65	35.07 ± 2.75
Average		7.67 ± 0.54	0.74 ± 0.09	62.92 ± 3.95	21.92 ± 2.41	43.71 ± 15.85

**Table 3 plants-11-01337-t003:** Flavonoid contents of fruit peel and flesh (mg per 100 g of dry weight) harvested in 2018.

		HD	NRT
Peel	Control	463.1 ± 1.11	60.8 ± 1.51
	*Jedae-unshiu*	546.6 ± 3.38 *	78.5 ± 0.61 *
Flesh	Control	178.8 ± 0.43	92.4 ± 0.59
	*Jedae-unshiu*	276.3 ± 2.07 *	132.2 ± 1.53 *

* indicates a value with a significant difference compared to the control (*p* < 0.05). HD: Hesperetin, NRT: Narirutin.

## Data Availability

Not applicable.

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
