# Peer review of "The Citrus Mutant Jedae-unshiu Induced by Gamma Irradiation Exhibits a Unique Fruit Shape and Increased Flavonoid Content"

_plants, 2022, doi:10.3390/plants11101337_

Round 1
Reviewer 1 Report
Research questions are well defined and within the aims and the scope of the journal. Material is accordingly defined. Methods are suitable, mainly properly described. The investigation is performed to good technical standards. It is no ethical problem involved. Conclusions are well stated and based on the results. Discussion is sound and relevant.
Suggestions:
In the text above the Table 2, instead of: »Sugar, Acidity,«, better »sugar, acidity,«
Explain the data on gamma irradiation.
Author Response
We appreciate for your review and suggestions/
[Response to reviewer’s suggestions]
In the text above the Table 2, instead of: »Sugar, Acidity,«, better »sugar, acidity,«
>> WE changed as “sugar, acidity”
Explain the data on gamma irradiation.
>> Previously, we performed the irradiation with gamma ray. So, we added the references and changed the section2.1 as follows;
We previously performed a gamma rays (60Co) irradiation onto C. unshiu Marc. cv. Miyagawa-wase scions and then grafted onto the branches of C. unshiu Marc. cv. Miya-gawa-wase trees to select mutant lines. Several mutant lines, such as a unique fruit shape, high sugar content, peel color, or late maturation etc., were selected that year. The next year, the scions of mutant lines selected were used for veneer-grafting onto rootstocks (C. unshiu Marc. cv. Miyagawa-wase) and the lines were maintained (Figure 1) [14, 15]. Of several mutant lines, Jedae-unshiu, showing a unique fruit shape, was selected for further analysis and investigated whether its traits were maintained over 3 years, from 2018 to 2020.
Reviewer 2 Report
The communication describes a new mutant of satsuma obtained by gamma irradiation. The mutation obtained is a corrugated fruit. This kind of mutation is relatively frequent on natural and induced citrus mutants. The authors emphasize different characteristic of the mutated fruit in terms of firmness and accumulation of phytochemicals. They also mention the potential of the mutant as a good storage variety, but they did not evaluated storage. However, firmess is not enough to guarantee good storage ability, so the note does not indicate if this mutant has really improved post-harvest characteristics.
The manuscript does not contain important information of the methods used and must be improved in the following parts:
line 35: eggplant? please explain. Rather, in this part you should add bibliographic information related to the parentage of satsuma, based on the recent release of its genome.
line 36: use “selection” instead of “strain”
lines 46-51: I suggest to eliminate this sentence. You are dealing with satsumas, so it's not worth it to write about oranges and grapefruit. Moreover, Star ruby was released decades ago, so it cannot be considered a new variety
line 56: satsuma seeds are polyembryonic, not polyploid
line 60: I would add "and mandarins".
Section 2.1: information should be added in this section. The authors do not specify the methods used for the irradiation, specifically doses and dose rates. Also, the authors do not specify the number of irradiated buds and the number of survived buds.
Please delete line 77-78
lines 89-92: please delete the comparison with other species, not needed. The authors could rather describe similar experiences in citrus
lines 98-100: statistical significance in table 1 is missing
line 123: what is Moroccan clementina? please check
table 3: please specify in which year you performed the HPLC experiment
line 145: control
methods: the authors do not specify the number of fruits they used to perform the analysis for the three years of investigation
Author Response
We appreciate for your review and suggestions/
[Response to reviewer’s comments]
They also mention the potential of the mutant as a good storage variety, but they did not evaluated storage. However, firmess is not enough to guarantee good storage ability, so the note does not indicate if this mutant has really improved post-harvest characteristics.
>> On the good storage ability, we just hope to be mentioned for the possibility. So, we changed from “potential” to “possibility” in the abstract
The manuscript does not contain important information of the methods used and must be improved in the following parts:
line 35: eggplant? please explain.
>> we changed as “a branch”
Rather, in this part you should add bibliographic information related to the parentage of satsuma, based on the recent release of its genome.
>> We hope to be described the origin story of C. unshiu. And we changed the sentence as “Miyagawa wase originated in Fukuoka Prefecture of Japan as a natural mutant of ordinary C. unshiu.”
line 36: use “selection” instead of “strain”
>> we changed as “strain”
lines 46-51: I suggest to eliminate this sentence. You are dealing with satsumas, so it's not worth it to write about oranges and grapefruit. Moreover, Star ruby was released decades ago, so it cannot be considered a new variety
>> we eliminated the sentence.
line 56: satsuma seeds are polyembryonic, not polyploid
>> we changed as “polyembryonic”
line 60: I would add "and mandarins".
>> we added "and mandarins".
Section 2.1: information should be added in this section. The authors do not specify the methods used for the irradiation, specifically doses and dose rates. Also, the authors do not specify the number of irradiated buds and the number of survived buds.
>> Previously, we performed the irradiation with gamma ray. So we added the references.
>> we changed the section2.1 as follows;
We previously performed a gamma rays (60Co) irradiation onto C. unshiu Marc. cv. Miyagawa-wase scions and then grafted onto the branches of C. unshiu Marc. cv. Miya-gawa-wase trees to select mutant lines. Several mutant lines, such as a unique fruit shape, high sugar content, peel color, or late maturation etc., were selected that year. The next year, the scions of mutant lines selected were used for veneer-grafting onto rootstocks (C. unshiu Marc. cv. Miyagawa-wase) and the lines were maintained (Figure 1) [14, 15]. Of several mutant lines, Jedae-unshiu, showing a unique fruit shape, was selected for further analysis and investigated whether its traits were maintained over 3 years, from 2018 to 2020.
Please delete line 77-78
>> we deleted it
lines 89-92: please delete the comparison with other species, not needed. The authors could rather describe similar experiences in citrus
>> Several studies on citrus mutants induced by gamma irradiation were manly focused on seedless phenotype, not the external phenotype of citrus fruit. So we described the discussion about non-citrus species
lines 98-100: statistical significance in table 1 is missing
>> we added the statistical significance
line 123: what is Moroccan clementina? please check
>> Oufedjikh et al. (1998) described as “Moroccan clementina”
table 3: please specify in which year you performed the HPLC experiment
>> we added the year “2018”
line 145: control
>> we changed as “control”
methods: the authors do not specify the number of fruits they used to perform the analysis for the three years of investigation
>> we described as “Jedae-unshiu and control were investigated by harvesting 10 or more fruits per tree from 5 or more trees each year.”
Round 2
Reviewer 2 Report
The authors still have to include the information requested and fix some parts:
-dose and dose rates used for gamma irradiation are still missing
- There are several reports indicating that mutation breeding have been used to improve other fruit characters in citrus, not just seedlessness. Therefore the authors should include reports in which gamma rays caused changes in the fruit phenotype. See for example the book "The Genus Citrus" to find examples.
- "Moroccan clementina" is not correct, even if it was described like this. It should be "Moroccan Clementine". Or ,better, "Citrus clementina", which is the species name
Author Response
We appreciate for your comments and changed our manuscript as follows
- There are several reports indicating that mutation breeding have been used to improve other fruit characters in citrus, not just seedlessness. Therefore the authors should include reports in which gamma rays caused changes in the fruit phenotype. See for example the book "The Genus Citrus" to find examples.
>> We added the dose and dose rates used for gamma irradiation
“We previously performed a gamma ray irradiation (60Co, 100Gy with 60% survival rate of budsticks) onto C. unshiu……”
- "Moroccan clementina" is not correct, even if it was described like this. It should be "Moroccan Clementine". Or ,better, "Citrus clementina", which is the species name
>> We changed as “Moroccan Clementine”